# An Atmospheric Phase Correction Method Based on Normal Vector Clustering Partition in Complicated Conditions for GB-SAR

Pengfei Ou [1], Tao Lai [1,*], Shisheng Huang [2], Wu Chen [3] and Duojie Weng [3]

1   School of Electronics and Communication Engineering, Shenzhen Campus, Sun Yat-sen University, Shenzhen 518107, China
2   Beijing Institute of Tracking and Communication Technology, Beijing 100094, China
3   Department of Land Surveying and Geo-Informatics, The Hong Kong Polytechnic University, Hong Kong 999077, China
*   Correspondence: lait3@mail.sysu.edu.cn

**Abstract:** Atmospheric phase is the main factor affecting the accuracy of ground-based synthetic aperture radar. The atmospheric phase screen (APS) may be very complicated, due to the drastic changes in atmospheric conditions, and the conventional correction methods based on regression models cannot fit and correct it effectively. Partition correction is a feasible path to improve atmospheric phase correction (APC) accuracy for complicated APS, but the overfitting problem cannot be ignored. In this article, we propose a clustering partition method, based on the normal vector of APS, which can partition the complicated APS more reasonably, and then perform APC based on the partition results. APC, and simulation experiments on measurement data, suggests that the proposed method achieves higher accuracy than the conventional model-based methods for complicated APS and avoids severe overfitting, realizing the balance between accuracy and credibility. This article verifies the feasibility and effectiveness of using APS distribution information to guide the partition and conduct APC.

**Keywords:** ground-based synthetic aperture radar (GB-SAR); atmospheric phase (AP); permanent scatterer (PS); complicated atmospheric condition; regression model; k-means; clustering partition

## 1. Introduction

Ground-based synthetic aperture radar (GB-SAR) interferometry, is a valid measurement technique for deformation and has been widely utilized in the fields of geological hazard monitoring and building deformation monitoring [1]. Fine spatial resolution, zero spatial baseline, and short revisiting time, make GB-SAR a suitable sensor for the monitoring of small-scale areas [2]. A GB-SAR system extracts the deformation component from the signal phase by an interferometry process, to achieve submillimeter precision deformation inversion. Compared with conventional techniques, such as displacement meter, GNSS, total station, and LIDAR, GB-SAR has significant advantages such as its ability to be used all day, in all weathers, having a high accuracy and wide spatial coverage, and fast image acquisition [3].

A spatially inhomogeneous atmosphere affects the speed and path of radar waves in propagation, introducing additional phase delays [4,5]. Because the atmospheric conditions constantly change over time, the phase delay in images are not consistent and cannot be eliminated by the interferometry process. The residual phase delay in the interferogram is known as atmospheric phase (AP), and it is one of the main factors affecting the precision of GB-SAR. Mainstream GB-SAR systems use short wavelength electromagnetic waves and are susceptible to changes in atmospheric conditions [6]. Atmospheric phase correction (APC) is one of the most critical challenges in obtaining high-accuracy and reliable deformation inversion results in GB-SAR.

The atmospheric refractive index, which affects radar waves, is determined by humidity, pressure, and temperature. Theoretically, phase delays can be calculated with the auxiliary of external meteorological data [7]. However, in real monitoring scenarios, it is quite difficult to obtain high-precision meteorological data. The most practical APC methods are currently based on the permanent scatterer (PS) technique. The scattering properties of PSs could remain highly stable in long-term monitoring, and their phase change is mainly caused by atmospheric changes [8]. By analyzing the phase of PSs, a variety of regression models without auxiliary data have been proposed to correct AP [9,10]. However, these models have low order and few parameters, and their fitting ability is poor. Current research works [2,10] and our dataset, reveal that the atmosphere changes drastically when the monitored target is located at a high altitude and has a steep topography, AP can not be accurately corrected in these complicated conditions by simple models. To further improve the APC accuracy, some data-driven APC methods have been proposed. These methods, discussed in [6,11–13], generally divide the atmospheric phase screen (APS) into a massive number of sub-blocks and correct them by interpolation or parameter estimation, to achieve high accuracy. However, APC in a small sub-block is prone to overfitting, which reduces the credibility of APC and deformation inversion.

Partition correction is a feasible path to improve the APC accuracy for complicated APS, but overfitting cannot be ignored. To solve the APC problem of GB-SAR in complicated atmospheric conditions, a new APC method, based on a clustering partition, is proposed in this article. Spatial normal vectors of the PSs are estimated and combined with position coordinates, to construct new vectors. Then, a clustering process is utilized to partition the APS into sub-blocks; finally, APC is performed in each sub-block. A dataset was acquired from a mine in Shaoguan City, Guangdong Province, with an Arc-SAR system, and was processed. The APS estimation on interferograms, a simulation experiment, time series of APC, and the comparisons with the conventional methods, validated the feasibility and effectiveness of the proposed method. Setting the algorithm parameters reasonably, the proposed method has effectively balanced the contradiction between accuracy and overfitting.

The arrangement of this article is as follows: Section 2 introduces some related works about APC; Section 3 introduces the information of the dataset and briefly analyzes it; Section 4 introduces the new method; Section 5 presents the results and analysis; Sections 6 and 7 present the discussion and conclusions, respectively.

## 2. Related Works

For a radar image acquired at time $t$, its unwrapped phase can be modeled as

$$\varphi(\vec{r}, t) = \varphi_0 + \frac{4\pi f_c}{c} r + \varphi_{atm}(\vec{r}, t) + \varphi_{noise} \tag{1}$$

where $\vec{r}$ denotes the space domain, $\phi_0$ is the backscattering phase, $f_c$ is the carrier frequency, $c$ is the speed of light, $r$ is the slant distance from the radar to target, $\varphi_{atm}(\vec{r}, t)$ is the AP delay for domain $\vec{r}$ at time $t$, and $\varphi_{noise}$ denotes the thermal noise [14].

The atmosphere is a non-ideal medium, that affects the speed and path of electromagnetic wave propagation. Let atmospheric refractive index be $n(\vec{r}, t)$, then the corresponding atmospheric delay $\varphi_{atm}(\vec{r}, t)$ can be obtained by integrating $n(\vec{r}, t)$ along the propagation path $L$, as is shown in Equation (2) [15].

$$\varphi_{atm}(\vec{r}, t) = \frac{4\pi f_c}{c} \int_L n(\vec{r}, t) dl \tag{2}$$

The AP term $\Delta\varphi_{atm}$ of an interferogram is described as

$$\Delta\varphi_{atm} = \frac{4\pi f_c}{c} \int_L \Delta n(\vec{r}, t) dl \tag{3}$$

where $\Delta n$ denotes the refractivity variation.

APC methods can be mainly divided into two categories: methods based on meteorological data and methods based on PSs. In recent years, some more novel methods have also emerged.

### 2.1. Methods Based on Meteorological Data

$\varphi_{atm}(\vec{r}, t)$ can be calculated by using meteorological data to calculate $\Delta n$. According to the ITU-R recommendation, the refractive index $n$, is determined by the following expression [16]:

$$n = 1 + 10^{-6} \cdot (77.6 \frac{P}{T} - 5.6 \frac{e}{T} + 3.75 \times 10^5 \frac{e}{T^2}) \tag{4}$$

where $T$ is absolute temperature (K), $e$ is water vapor pressure (hPa), and $P$ is total atmospheric pressure (hPa). $e$ can be calculated using $T$, $P$, and relative humidity $H$ (%).

Luzi et al. [17] used meteorological data to compensate for AP, and concluded that meteorological data at a single location could not be used to correct for AP at different ranges. Iannini L. et al. [18] used meteorological data to model tropospheric atmospheric refraction and successfully rejected most of AP, confirming the feasibility of the meteorological data APC method.

The meteorological data-based method is theoretically supported, but its accuracy is limited by the coverage and accuracy of meteorological sensors, which results in its poor practicability.

### 2.2. Methods Based on Permanent Scatterers

#### 2.2.1. Permanent Scatterers

The PS technique was initially applied in satellite-based SAR [19], and later extended to GB-SAR. PSs have stable scattering characteristics and can be assumed to be stationary, so that the AP can be inferred from the phase change of PSs. PSs-based APC methods are most widely used. The most commonly used criteria for PSs selection include amplitude-dispersion index (ADI) and coherence coefficient.

The ADI method was proposed by Ferretti [19], it is simple and efficient, but requires a large number of SAR images for statistics. The expression for calculating the ADI $D_A$ is

$$D_A = \frac{\sigma_A}{m_A} \tag{5}$$

where $\sigma_A$ is the standard deviation and $m_A$ is the mean value of a pixel's time series.

The coherence coefficient method was proposed by Berardino et al. [20]. A minimum of two images is required to select PS points using the coherence coefficient method, but isolated highly coherent points are easily missed. The coherence coefficient, $\gamma$, of a pixel's time series is calculated as

$$\gamma = \frac{|\sum\limits_{i=1}^{m} \sum\limits_{j=1}^{n} M(i,j)S^*(i,j)|}{\sqrt{\sum\limits_{i=1}^{m} \sum\limits_{j=1}^{n} |M(i,j)|^2 \sum\limits_{i=1}^{m} \sum\limits_{j=1}^{n} |S(i,j)|^2}} \tag{6}$$

where $M(i,j)$ is the master image and $S(i,j)$ is the slave image of an interferogram pair, $*$ denotes the complex conjugate, and $m \times n$ represents the size of the sliding window [21]. The average coherence coefficient of a pixel $\bar{\gamma}$ is

$$\bar{\gamma} = \frac{\sum\limits_{i=1}^{N} \gamma_i}{N} \tag{7}$$

where $\gamma_i$ is the coherence coefficient of the $i$th interferogram pair of the pixel, and $N$ is the number of interferogram pairs.

Phase deviation and phase noise are also used in PSs selection. Generally, PSs selection uses a combination of criteria, to improve selection quality.

### 2.2.2. Conventional Model-Based Methods

In a stable atmosphere, $T$, $H$, and $P$ vary slowly in time and are homogeneous in space. $\Delta n$ can be considered as a constant. Therefore, Equation (3) can be summarized as the following slant distance regression model

$$\Delta\varphi_{atm} = \frac{4\pi f_c \Delta n}{c} \int_L dl = \beta \cdot r \tag{8}$$

where $r$ is the radar-to-target distance and $\beta$ is the regression coefficient. For further fitting ability, the 2nd-order model shown in Equation (9) [22], or higher-order models, have also been utilized in GB-SAR applications.

$$\Delta\varphi_{atm} = \beta_0 + \beta_1 \cdot r + \beta_2 \cdot r^2 \tag{9}$$

where $\beta_0$, $\beta_1$, and $\beta_2$ are the model regression coefficients.

For complicated scenarios, with steep topographic variations, multiple-regression models are proposed. In these scenarios, the atmosphere can be considered as stratified in height, and $n$ shows an exponential relationship with height, whereby the slant distance and elevation model can be utilized to estimate the APS, as is shown in the following equation [9]:

$$\Delta\varphi_{atm} = \beta_0 + \beta_1 \cdot r + \beta_2 \cdot r \cdot \Delta h \tag{10}$$

where $\beta_0$, $\beta_1$, and $\beta_2$ are the model regression coefficients; and $\Delta h$ denotes the relative elevation between the radar and the target. The acquisition of $\Delta h$ requires auxiliary data such as an accurate DEM, which is difficult in emergencies, limiting the application of the slant distance and elevation model. Based on [9], Dematteis [23] proposed a new model:

$$\Delta\varphi_{atm} = \beta_0 + \beta_1 \cdot r + \beta_2 \cdot r \cdot h^2 \tag{11}$$

where $h$ is elevation.

In scenarios with wide span, the AP changes significantly in both range direction and azimuth direction. The introduction of azimuth angle can effectively improve the fitting ability [24]. The slant distance & azimuth model is shown in Equation (12):

$$\varphi = \frac{4\pi}{\lambda}(\beta_0 + \beta_1 \cdot r + \beta_2 \cdot \sin\theta) \tag{12}$$

where $\theta$ is the azimuth angle.

When the monitored target spans a large distance, the interferograms can be partitioned in the range or azimuth direction, and APC is performed separately for each sub-block to enhance the correction effect.

### 2.2.3. Some Novel Data-Driven Methods

Model-based methods are not applicable for complicated conditions, some novel methods based on the properties of the APS are proposed. These methods can better simulate AP and have higher APC accuracy. Based on the path integral model of AP theory, Yunkai Deng et al. [11] proposed a grid partition method, this method firstly divides the interferogram into a certain number of grids, and then estimates the atmospheric parameters of the grid to achieve APC; Cheng Hu [12] and Xiaolong Zhao et al. [6] selected appropriate reference control PSs, to reconstruct the APS using spatial interpolation. Such fine methods generally divide the APS into hundreds of sub-blocks and correct them by interpolation or parameter estimation, to obtain high-accuracy APC results. However,

distinguishing the potential deformation phase component from the AP in a small sub-block is challenging, the estimation of the APS is prone to overfitting, which reduces the credibility of the APC and deformation inversion results. Yuta Izumi et al. [13] performed clustering, in accordance with the temporal APS behavior, to divide sub-blocks; the number of sub-blocks and the effect of overfitting are reduced. However, time-series clustering requires a large amount of previous monitoring images to be included in the data processing, this method is computationally intensive and has poor real-time performance.

## 3. Data Acquisition and Analysis

### 3.1. Data Acquisition

The dataset used in this article was acquired by an Arc-SAR system, as is shown in Figure 1b. The main system parameters are shown in Table 1.

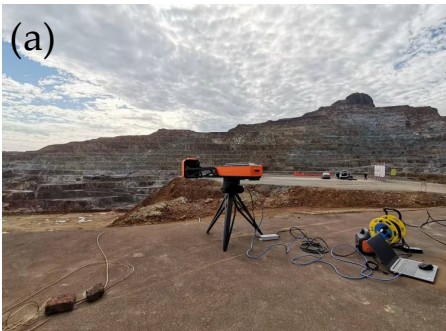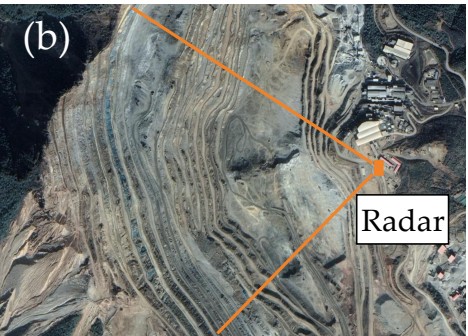

**Figure 1.** Main mine of Dabao Mountain Mine, the Arc-SAR system and its location: (**a**) main mine and the Arc-SAR system, (**b**) location of the Arc-SAR system.

**Table 1.** System parameters of the Arc-SAR system.

| Parameters | Value |
| :---: | :---: |
| Carrier frequency | 24 GHz |
| Beam coverage | 270° |
| Range resolution | 0.2 m |
| Angle resolution | 6 mrad |
| Detection range | 4000 m |

The monitored target is the main mine of Dabao Mountain Mine, in Shaoguan City, Guangdong Province, China, as is shown in Figure 1a. The elevation of the Dabao Mountain Mine is from 600–800 m. After decades of open-pit mining, the mountain is stepped and fairly steep. There is a large pit in the foot of the mine, with the risk of landslide collapse. The radar was located on the east side of the mine, about 800 m away from the mountain's main body, and can cover the whole mountain, as is shown in Figure 1b. The data acquisition campaign started at 11:00 a.m. on 10 November 2021, and ended at 11:00 a.m. on 12 November, with 849 SAR images obtained. During the data acquisition, the mine was in the normal mining condition.

### 3.2. Data Analysis

#### 3.2.1. PSs Selection

Figure 2 shows the PSs selection result, using ADI and the coherence coefficient as the phase quality metrics. By setting an ADI threshold $D_{th} = 0.1$ and an average coherence coefficient threshold $\gamma_{th} = 0.95$, 147,624 PS points are selected in the first 100 interferograms.

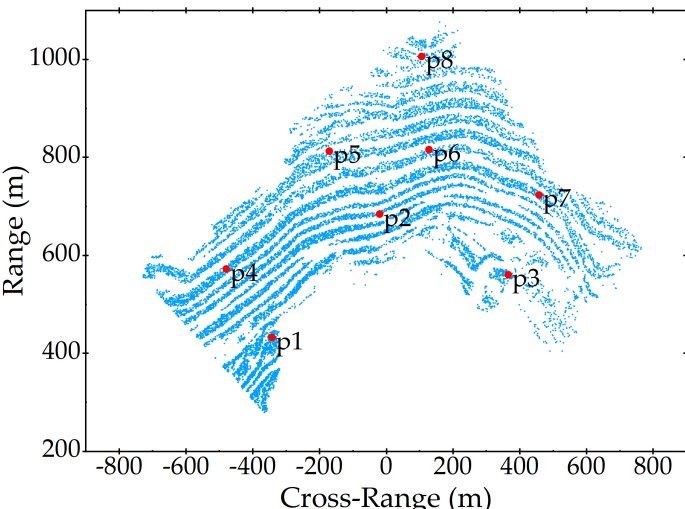

**Figure 2.** Permanent scatterers (PSs) selection result, $p_1$–$p_8$ are the reference PS (RPS) points for phase analysis.

### 3.2.2. Spatial Phase Wrapping in the Dataset

In the continuous mode, when the temporal baseline is short, phase wrapping usually appears only in the time dimension, and it is only in the discontinuous monitoring mode that the spatial phase wrapping needs to be considered [25,26]. The acquisition interval of the radar images was less than 3 min, but a significant spatial phase wrapping phenomenon appeared. Figure 3 shows one of the interferograms with spatial phase wrapping. This phenomenon suggests that the possibility of two-dimensional spatial phase wrapping cannot be ignored in the continuous observation mode with a short temporal baseline, when the atmospheric conditions are complicated. In GB-SAR, there is no complicated interference fringe when phase wrapping occurs, and the coherence between the two images is also strong, which makes the phase unwrapping simpler than airborne SAR and spaceborne SAR. In our article, based on the distribution of residues, a wrapped interferogram is divided into high quality regions and low quality regions. For the high quality areas, phase unwrapping is performed by integration along the range and azimuthal direction; for the low quality areas, the branch-cut algorithm [25,27] is used. The experimental results show that the idea greatly improves the efficiency, while ensuring the quality of unwrapping.

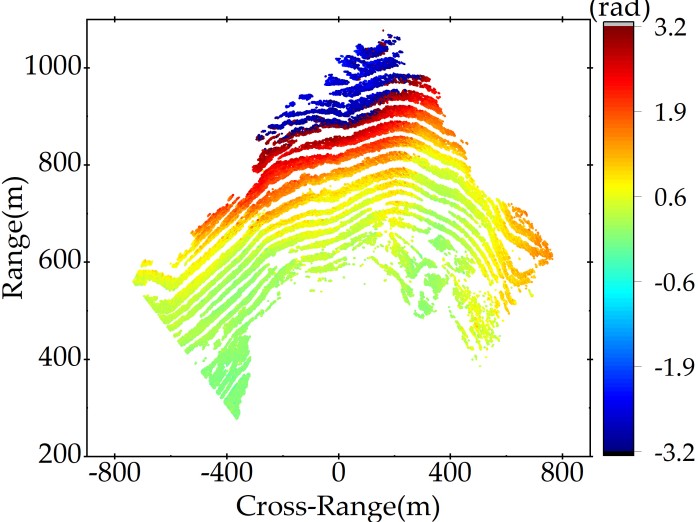

**Figure 3.** An interferogram with spatial phase wrapping, only PSs are displayed.

### 3.2.3. Cumulative Phase Analysis

Dividing the image into two cross-range intervals and four range intervals, eight blocks were obtained. The pixel with the lowest ADI in each block was taken as the reference PS (RPS) point, their distribution is marked in Figure 2. The uncorrected cumulative phase (CP) curves of the RPS points are shown in Figure 4. For PSs in each interferogram, the standard deviation of their phase values was calculated, Figure 5 shows the curve of the standard deviation. The larger the standard deviation, the more complicated the APS.

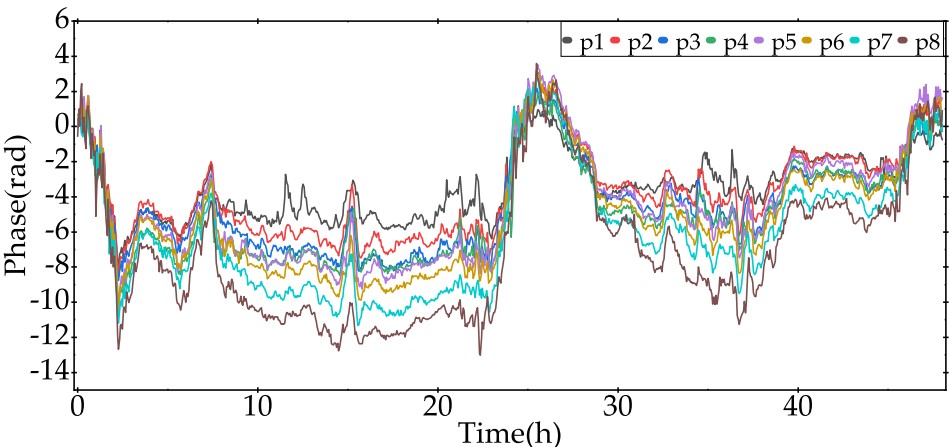

**Figure 4.** Cumulative phase (CP) curves of the RPS points.

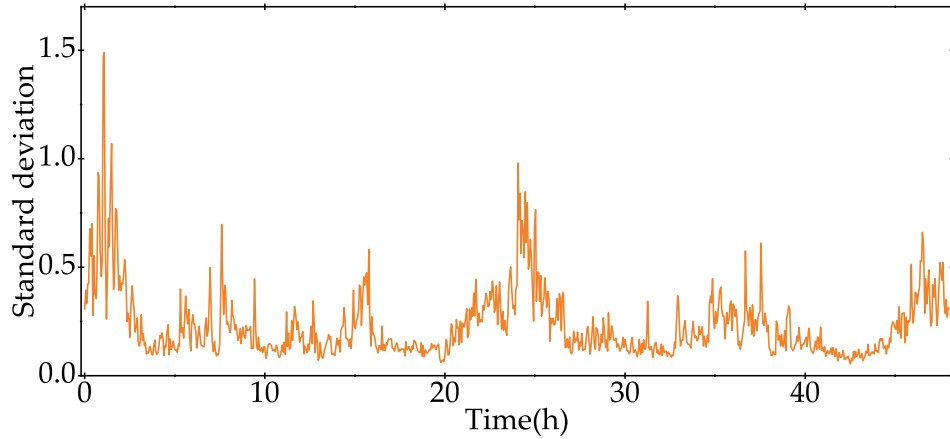

**Figure 5.** Phase standard deviation curve of all interferograms, only PSs counted.

A significant periodicity is observed in the changes of the CP curves in Figure 4 and the standard deviation curve in Figure 5. In the morning, the CP curves show a clear increasing tendency and the standard deviation of the interferograms also increased, reaching a maximum value at around 14:00. Then, the CP curve starts to decline, and the standard deviation of the interferogram decreases and stabilizes in the evening. The change in the CP curves at night is flatter than that of the daytime, and the standard deviation of the interferometric phase is also smaller.

## 4. Methodology

### 4.1. Algorithm Flowchart

The workflow of the proposed method is shown in Figure 6. The main steps include data pre-processing, normal vector estimation, clustering partition, and atmospheric phase correction.

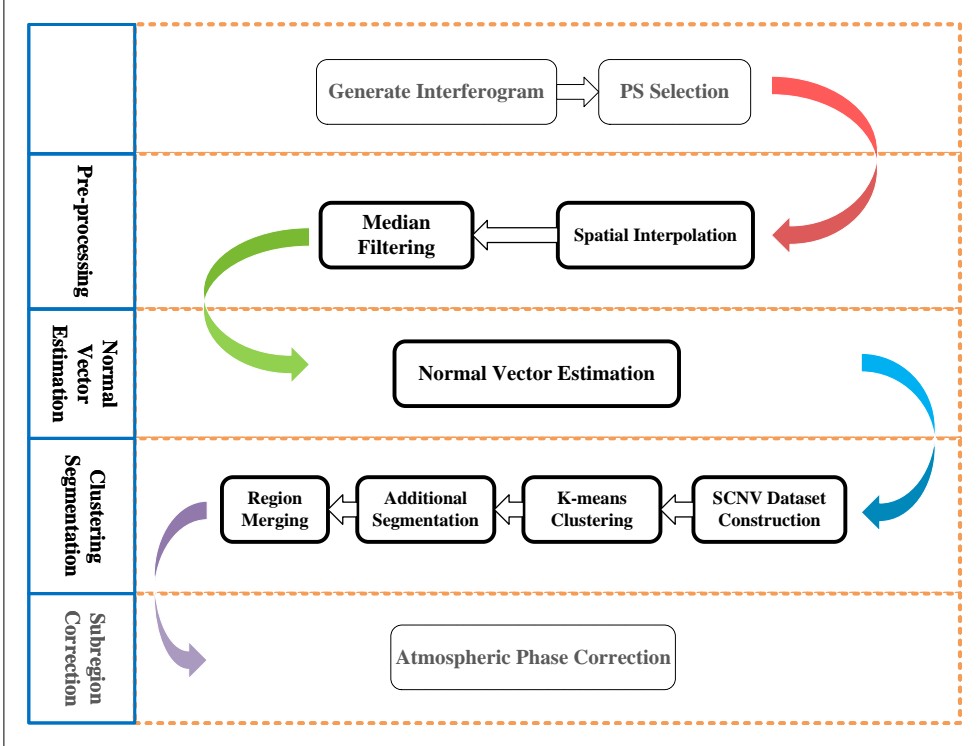

**Figure 6.** Workflow of the proposed method.

*4.2. Data Pre-Processing*

The differences in the stability of the monitored target's scattering properties lead to a non-uniform distribution of PSs. When AP estimation is performed in sub-blocks, the influence of the dense areas of PSs on the estimation results, is weighted more than that of the sparse areas of PSs, which biases the estimation. Therefore, the areas with sparse PSs need to be properly supplemented. Considering that the AP is highly relevant in local space, an inverse distance weighting interpolation (IDWI) algorithm, based on Delaunay triangulation, is utilized in the proposed method, to perform spatial interpolation. The main steps of spatial interpolation are:

1. Construct a Delaunay triangulation network, which is denoted $T$, to connect all PS points. The IDWI is performed only within the convex packet of PSs. Delaunay triangulation is an optimized spatial structure and can make the interpolation result automatically approach the regular triangle, improving the interpolation precision [28,29];

2. For a pending interpolation point $p$, in the convex package, it must be inside a triangle $T$, and three vertices of this triangle $v_1$, $v_2$, and $v_3$ are selected as reference points for interpolation;

3. The estimated phase of $p$, which is denoted $\varphi_p$, is calculated by

$$\varphi_p = \sum_{i=1}^{n} \frac{\varphi_{vi}}{|d_i|^2} \Big/ \sum_{i=1}^{n} \frac{1}{|d_i|^2} \tag{13}$$

where $i = 1, 2, 3$, $\varphi_{vi}$ are the phase values of reference points $v_1$, $v_2$, and $v_3$, and $d_i$ is the spatial distance from $v_i$ to $p$, respectively.

The purpose of spatial interpolation is to avoid excessive disparity in the distribution density of PSs, there is no need to pursue absolute uniformity. The original PSs and the supplementary PSs are collectively called the complete PSs (CPSs). After spatial interpolation, the CPSs are also smoothed by median filtering, to reduce the negative effect of noise on the subsequent normal vector estimation and obtain the overall distribution of the APS. An example of the original PSs and the corresponding CPSs is shown in Figure 7.

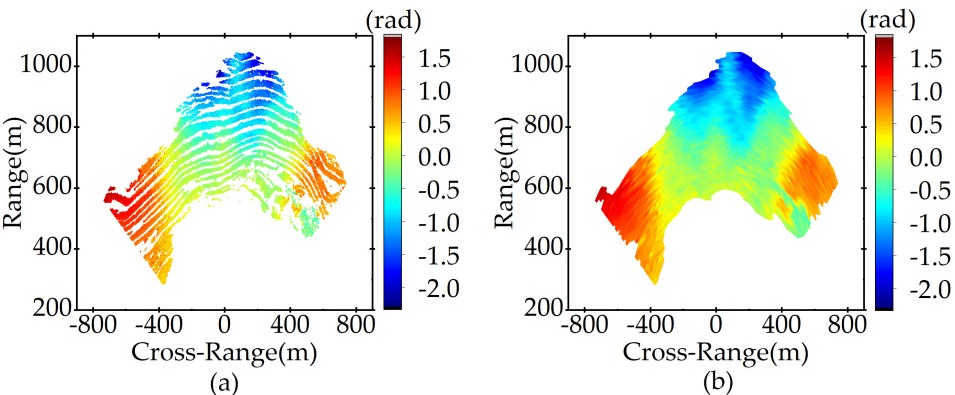

**Figure 7.** Distribution of the original PSs and the corresponding complete PSs (CPSs) after pre-processing: (**a**) original PSs, (**b**) CPSs.

### 4.3. Spatial Normal Vector Estimation

The space, composed of cross-range, range, and interferometric phase, which is denoted $\Omega$, is shown in Figure 8. The spatial span of CPSs is at the kilometer level, and the interferometric phase values are mostly below 10 rad, the spatial coordinate values of CPSs are not in the same order of magnitude as the phase values. If the normal vectors are estimated directly using the original phases, their directions will be too concentrated and will be difficult to be distinguished. Therefore, the interferometric phase is amplified, to ensure that its value range is of the same order of magnitude as the spatial span of the CPSs.

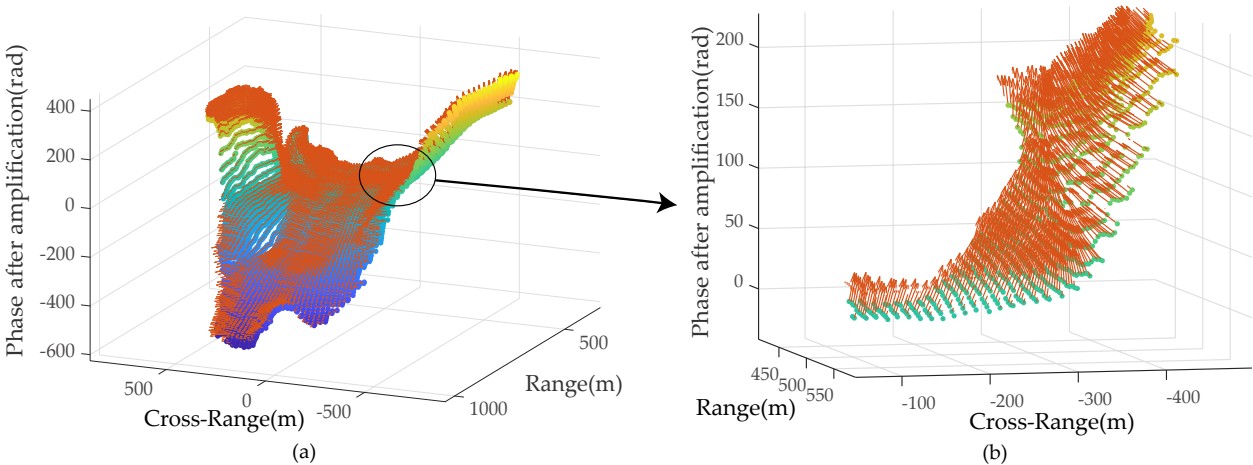

**Figure 8.** Schematic diagram of normal vector estimation: (**a**) normal vector estimation results of an amplified phase map and (**b**) local details. The orange arrows represent normal vectors.

The idea of estimating a normal vector, is to fit a plane based on the CPSs in the local APS, then the direction vector of the plane is the normal vector of a CPS. For a CPS $p(R\cos\theta, R\sin\theta, k_{ph}\varphi)$ in $\Omega$ (where $R$ is slant distance, $\theta$ is azimuth angle, $\varphi$ is phase, and $k_{ph}$ is amplification factor of $\varphi$), its normal vector $\vec{n}_p = (n_{CR}, n_R, n_P)^T$ is estimated by:

1.  Perform a k-nearest neighbor search, to find the nearest $k$ CPSs to $p$, the points set is denoted $N$:

$$N = \{p_i(x_i, y_i, z_i) | i = 1, 2, \ldots, k\} \tag{14}$$

where

$$\begin{cases} x_i = R_i \cos\theta_i \\ y_i = R_i \sin\theta_i \\ z_i = k_{ph}\varphi_i \end{cases} \tag{15}$$

where $R_i$, $\theta_i$, and $\varphi_i$ are slant distance, azimuth angle, and amplified phase of the $i$th CPSs in $N$, respectively.

The general form of the plane to be fitted is:

$$Ax + By + Cz + D = 0 \quad s.t. \quad A^2 + B^2 + C^2 = 1 \tag{16}$$

2.  Construct the covariance matrix $M$ for $N$:

$$M = \begin{bmatrix} \overline{x^2} - \bar{x}^2 & \overline{xy} - \bar{x}\bar{y} & \overline{xz} - \bar{x}\bar{z} \\ \overline{xy} - \bar{x}\bar{y} & \overline{y^2} - \bar{y}^2 & \overline{yz} - \bar{y}\bar{z} \\ \overline{xz} - \bar{x}\bar{z} & \overline{yz} - \bar{y}\bar{z} & \overline{z^2} - \bar{z}^2 \end{bmatrix} \tag{17}$$

where $\bar{x} = \frac{1}{k}\sum_{i=1}^{k} x_i$, $\overline{xy} = \frac{1}{k}\sum_{i=1}^{k} x_i y_i$, and so on.

3.  Solve the normalized eigenvectors corresponding to the minimum eigenvalue of $M$, to obtain the normal vector:

$$\vec{n_p} = \left(n_{CR}, n_R, n_\varphi\right)^T = \pm(A, B, C)^T \tag{18}$$

4.  Adjust the direction of $\vec{n_p}$ so that $(0,0,1) \cdot \vec{n_p} > 0$, to ensure that all normal vectors point to the same side.

### 4.4. Clustering Partition

The k-means clustering algorithm [30] is utilized in this article ,to divide the APS into an appropriate number of sub-blocks. Define the spatially constrained normal vector (SCNV) as

$$V_{scnv} = (R\cos\theta, R\sin\theta, k_{nv} \cdot n_{CR}, k_{nv} \cdot n_C, k_{nv} \cdot n_\varphi) \tag{19}$$

The SCNV of a CPS has two components: the position coordinates component $(R\cos\theta, R\sin\theta)$ and the normal vector component $(k_{nv} \cdot n_{CR}, k_{nv} \cdot n_C, k_{nv} \cdot n_\varphi)$, where $R$ and $\theta$ denote the slant distance and azimuth angle of the CPSs, respectively; $k_{nv}$ is the amplification factor of $(n_{CR}, n_R, n_\varphi)$. The value range of $(R\cos\theta, R\sin\theta)$ is usually much larger in order of magnitude than the norm of $(n_{CR}, n_R, n_\varphi)$. Therefore, the normal vector component needs to be amplified to raise its weight in SCNV. The empirical criterion for the selection of $k_{nv}$ is that the norm of the amplified normal vector is in the same order of magnitude as the value range of $(n_{CR}, n_R, n_\varphi)$. For the same interferogram, $k_{nv}$ is kept consistent.

Clustering partition undergoes the following steps [31]:

1.  Construct the SCNV set $V = \{V_{scnv}^1, V_{scnv}^2, \dots, V_{scnv}^m\}$, where $m$ is the number of all CPS points;
2.  Set the number of clusters to $k_{cl}$ and initialize the clustering center $C = \{C_1, C_2, \dots, C_{k_{cl}}\}$;
3.  Calculate the Euclidean distance from $V_{scnv}^i$ to each cluster center and assign $V_{scnv}^i$ to the cluster with the closest Euclidean distance;
4.  Calculate the center of mass of each cluster and update the cluster center with the center of mass;
5.  The above steps are iteratively processed until the clustering centers no longer change, or a predetermined number of iterations is reached.

$k_{cl}$ is related to the properties of the APS: when the APS is gentle, $k_{cl}$ takes a small value, and when the APS fluctuates a lot and its spatial distribution is complicated, a larger $k_{cl}$ is needed.

Figure 9 shows the clustering results of the datasets constructed with position coordinates $(R\cos\theta, R\sin\theta)$, normal vectors $(n_{CR}, n_R, n_\varphi)$, and SCNV. When only position coordinates are used for clustering, the sub-blocks are divided entirely based on the spatial distance, which does not reflect the spatial distribution of APS. If only normal vectors are used, without introducing position coordinates, two sub-blocks that are far apart will be

clustered into one cluster, while two sub-blocks that are close to each other, but slightly different in orientation, will be divided into different clusters. Small fluctuations in APS have a great impact on the clustering process, and the final clustering results are very fragmented, which is contrary to our original intention of dividing local APS according to their overall trends. When using the SCNV dataset for clustering, component $(n_{CR}, n_R, n_\varphi)$ restricts the orientation and trend of the clusters, and component $(R\cos\theta, R\sin\theta)$ restricts the position of the clusters.

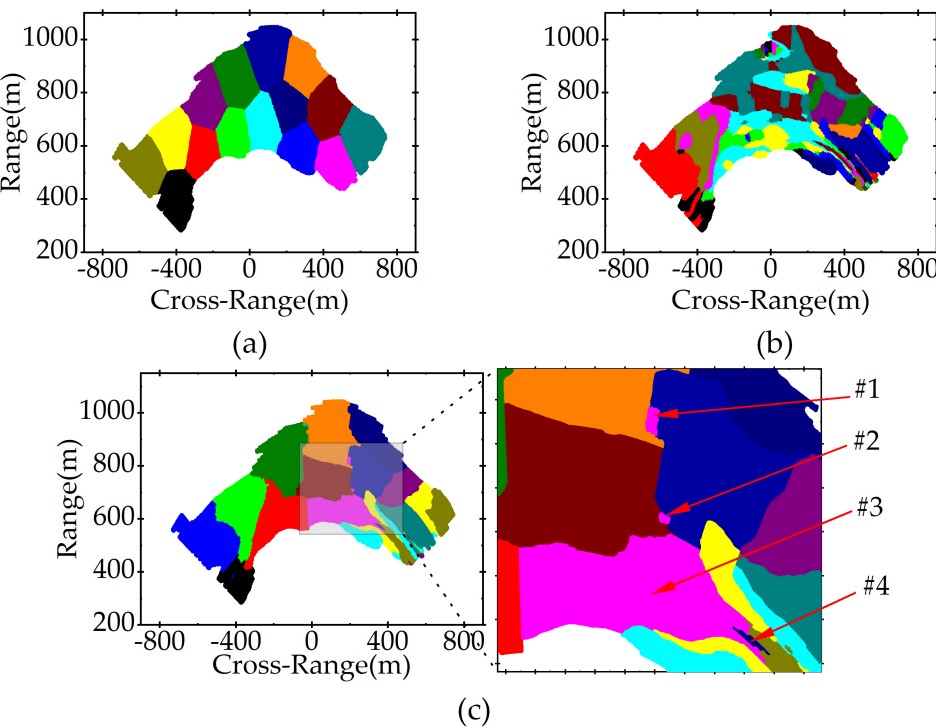

**Figure 9.** Clustering partition results using (**a**) position coordinate, (**b**) normal vector, and (**c**) spatially constrained normal vector. Different colors in the figures represent different clusters.

Although SCNV introduces the restriction of position coordinates, it does not completely ensure that CPSs clustered in the same cluster are necessarily spatially adjacent. Two sub-blocks with different positions and different normal vector orientations may be clustered into one cluster, when the normal vector components and the position components satisfy a specific relationship. For example, in Figure 9c, #1, #2, and #3 belong to the same cluster but are not spatially adjacent . Therefore, a secondary partition is needed, to divide the blocks distributed at different locations in the same cluster into several different sub-blocks.

After k-means clustering and secondary partition, there are a large amount of small sub-blocks, that are prone to overfitting in the APC process. These sub-blocks need to be merged with other larger sub-blocks. Defining the sub-blocks whose area is less than a certain threshold as too-small sub-blocks, the steps for merging a too-small sub-block are:

1. Calculate the average normal vector of each too-small sub-block;
2. Search the sub-blocks adjacent to the too-small sub-block, calculate their average normal vectors, and then calculate the spatial distance between these average normal vectors and the normal vector of the too-small sub-block;
3. Merge the too-small sub-block into the block corresponding to the minimum spatial distance in step 2.

### 4.5. Atmospheric Phase Correction

APS estimation and correction are performed in each sub-block, after the clustering partition. A schematic diagram of the process is shown in Figure 10.

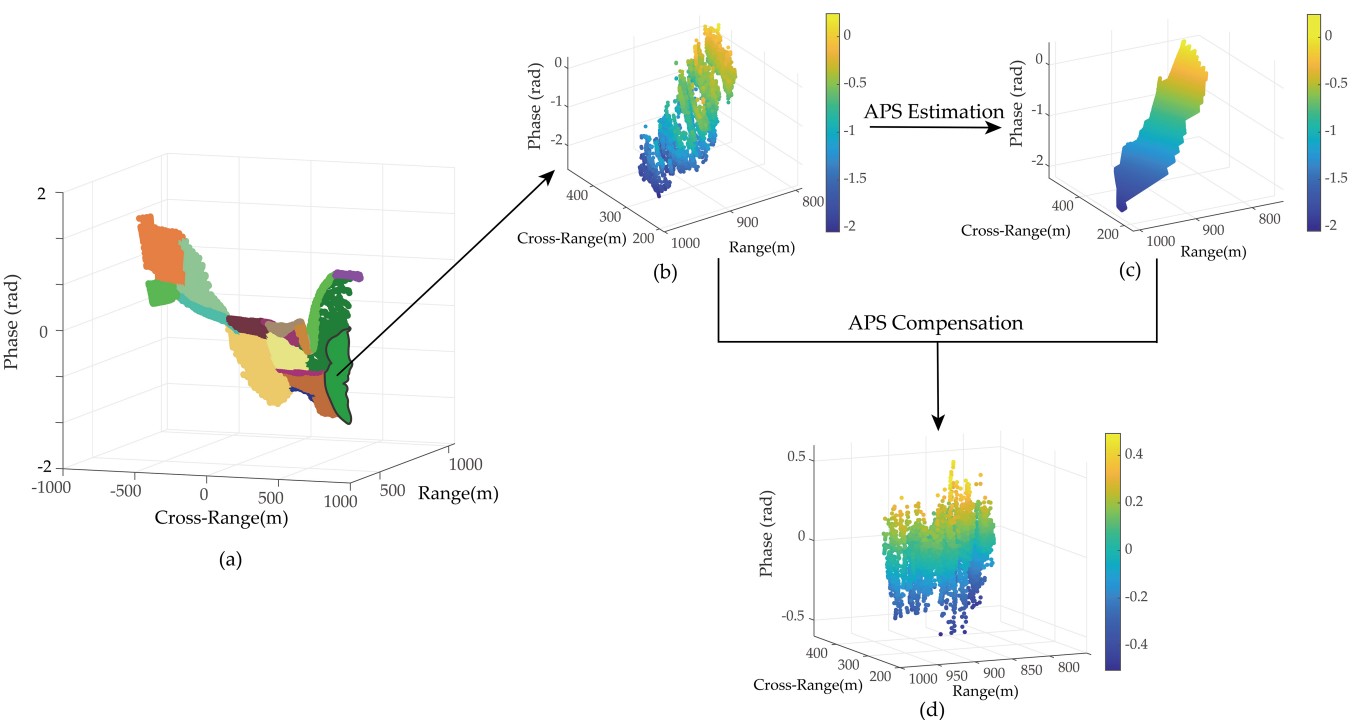

**Figure 10.** Schematic diagram of atmospheric phase screen (APS) estimation and correction: (**a**) APS partition results, different colors represent different sub-blocks; (**b**) a sub-block of APS; (**c**) APS estimation result of the sub-block in (**b**); (**d**) corrected result of the sub-block in (**b**).

The AP of each sub-block is estimated by the regression model shown in Equation (20).

$$\Delta \varphi_{atm} = \beta_0 + \beta_1 \cdot r \cdot \sin \theta + \beta_2 \cdot r \cdot \cos \theta \qquad (20)$$

The estimation of the regression coefficients $\beta_0 - \beta_2$ is performed based on CPSs. The matrix form of Equation (20) is

$$\Delta \mathbf{\Phi} = \mathbf{X}\beta + \varepsilon \qquad (21)$$

where

$$\Delta \mathbf{\Phi} = \begin{bmatrix} \Delta \varphi_1 \\ \Delta \varphi_2 \\ \cdots \\ \Delta \varphi_m \end{bmatrix}, \mathbf{X} = \begin{bmatrix} 1 & r_1 \cdot \sin \theta_1 & r_1 \cdot \cos \theta_1 \\ 1 & r_2 \cdot \sin \theta_2 & r_2 \cdot \cos \theta_2 \\ \cdots & \cdots & \cdots \\ 1 & r_m \cdot \sin \theta_m & r_m \cdot \cos \theta_m \end{bmatrix}, \beta = \begin{bmatrix} \beta_0 \\ \beta_1 \\ \beta_2 \end{bmatrix}, \varepsilon = \begin{bmatrix} \varepsilon_1 \\ \varepsilon_2 \\ \cdots \\ \varepsilon_m \end{bmatrix}, \quad (22)$$

where $m$ denotes the number of CPS points; $\Delta \varphi_i$, $r_i$ and $\theta_i$ are the unwrapped phases, slant distance, and azimuth of the $i$th CPS in the sub-block; $\beta_0 - \beta_2$ are the regression coefficients; and $\varepsilon$ is the random error vector. The regression coefficients vector can be estimated with a least-squares regression

$$\widehat{\beta} = (\mathbf{X}^T \mathbf{X})^{-1} \mathbf{X}^T \Delta \mathbf{\Phi} \qquad (23)$$

where $^T$ denotes the matrix transposition.

The estimated AP $\Delta \widehat{\mathbf{\Phi}}_{AP}$ is obtained from

$$\Delta \widehat{\mathbf{\Phi}}_{AP} = \mathbf{X}\widehat{\beta} \qquad (24)$$

The difference between $\mathbf{\Delta\Phi}$ and $\mathbf{\Delta\widehat{\Phi}}_{AP}$ is the corrected phase. The complete correction result is obtained after the correction of all sub-blocks.

## 5. Results and Analysis

### 5.1. APS Estimation on Interferograms

All interferograms were corrected by the 2nd-order slant distance model, the 2nd-order slant distance model with azimuthal partition, the slant distance & azimuth model, and the proposed method. The algorithm parameters of the proposed method are shown in Table 2.

**Table 2.** Algorithm parameters of the proposed method.

| Parameters | Value |
|:---:|:---:|
| $k_{ph}$ | 50 |
| $k_{cl}$ | 10 |
| $k_{nv}$ | 100 |

Since the monitored target was relatively stable and the probability of a large-scale landslide was small, the corrected phase should be concentrated around 0. Figure 11a shows the overall original phase distribution of the 7th interferogram's CPSs, and Figure 11b,c show the original phase distribution along the range direction and cross-range direction. In the range direction, the phase increases with distance, but the distribution in the cross-range direction is more complicated. The AP of the 7th interferogram is large, and complicated enough to verify the proposed method.

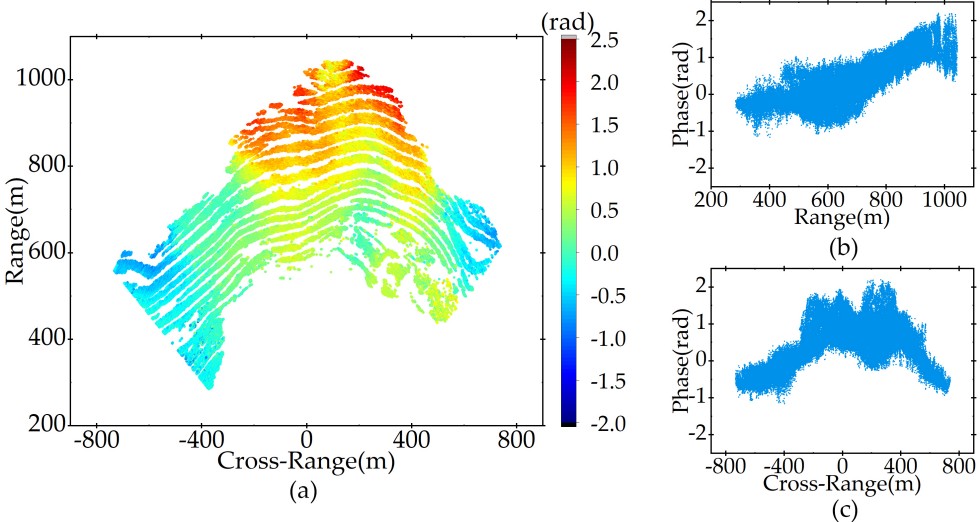

**Figure 11.** Original phase distribution of the 7th interferogram: (**a**) overall phase distribution, (**b**) phase distribution along the range direction, and (**c**) phase distribution along the cross-range direction.

Figure 12 shows the APC results by the three regression model-based methods and the distribution of the corrected phase along the different directions. The AP are all reduced after correction, and the improvement is particularly noticeable in the range direction. However, in general, there is still significant residual AP.

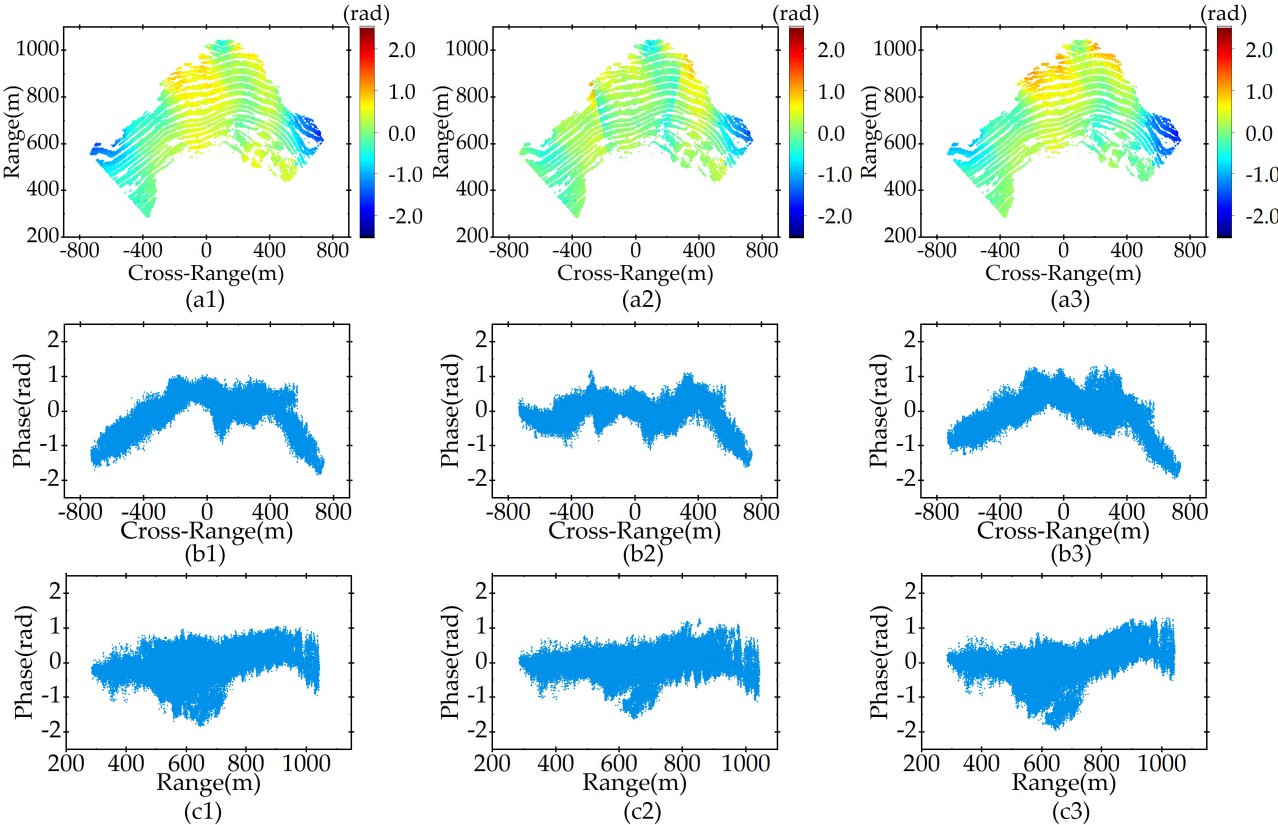

**Figure 12.** Correction results by the three regression model-based methods and the distribution of the corrected phases along the different directions: (**a1**–**a3**) are the atmospheric phase correction (APC) results of the 2nd-order slant-range model, the 2nd-order slant-range model with azimuthal partition, and the slant distance & azimuth model, respectively, (**b1**–**b3**) are the distributions of the corrected phase along the cross-range direction, and (**c1**–**c3**) are the distributions of the corrected phase along the range direction.

Figure 13 shows the APC results by the proposed method. High-accuracy results are obtained in both the range direction and cross-range direction, and the residual phase is mostly concentrated around 0 and significantly smaller than that of the regression model-based methods. Figure 14 shows the partition result of the 7th interferogram using the proposed method. The clustering partition result is consistent with the spatial distribution of APS, and the number of sub-blocks is less than 20.

Table 3 shows the statistical results of the residual AP's standard deviation after APC, and Figure 15 shows the variation curves of the residual AP's standard deviation after APC, along the acquisition time. From Table 3 and Figure 15, it is seen that the standard deviation of the corrected AP is significantly reduced compared with the uncorrected AP. The residual AP standard deviation of the slant-range model is the largest, and the fluctuation of its curves is the most obvious, but the fluctuation is significantly reduced after azimuthal partition. The smallest residual AP is obtained by the proposed method, with the smallest mean and median of the standard deviation and the smoothest curves, which indicates the highest APC accuracy.

Parameter setting analysis is conducted in Sections 5.3.1 and 5.3.3, to further explore the effect of the algorithm parameters on the corrected accuracy.

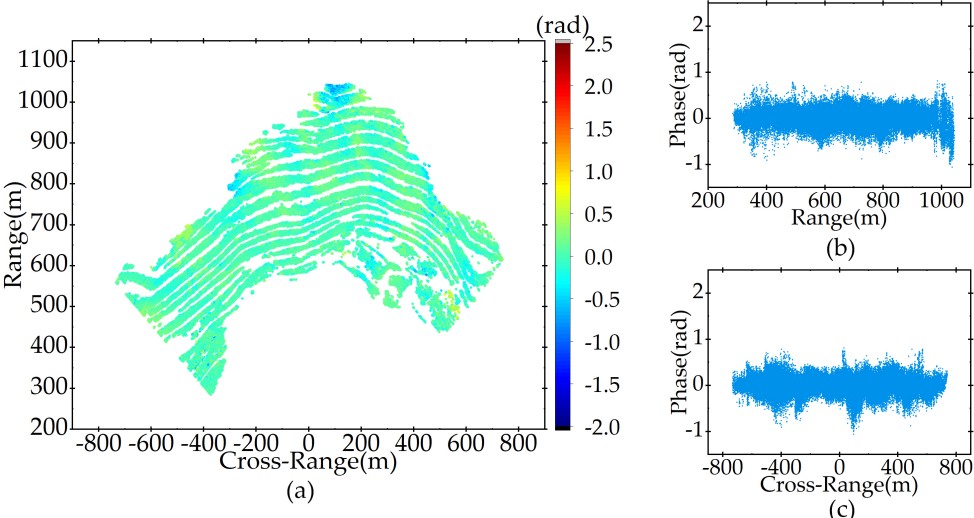

**Figure 13.** Corrected phase distribution of the 7th interferogram by the proposed method: (**a**) overall phase distribution, (**b**) phase distribution along the range direction, (**c**) phase distribution along the cross-range direction.

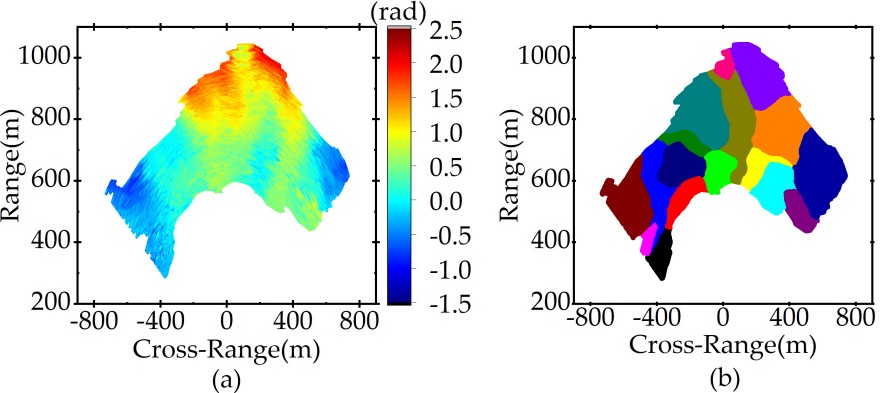

**Figure 14.** Clustering partition result of the 7th interferogram by the proposed method: (**a**) pre-processed result, (**b**) partition result. Different colors in (**b**) represent different clusters.

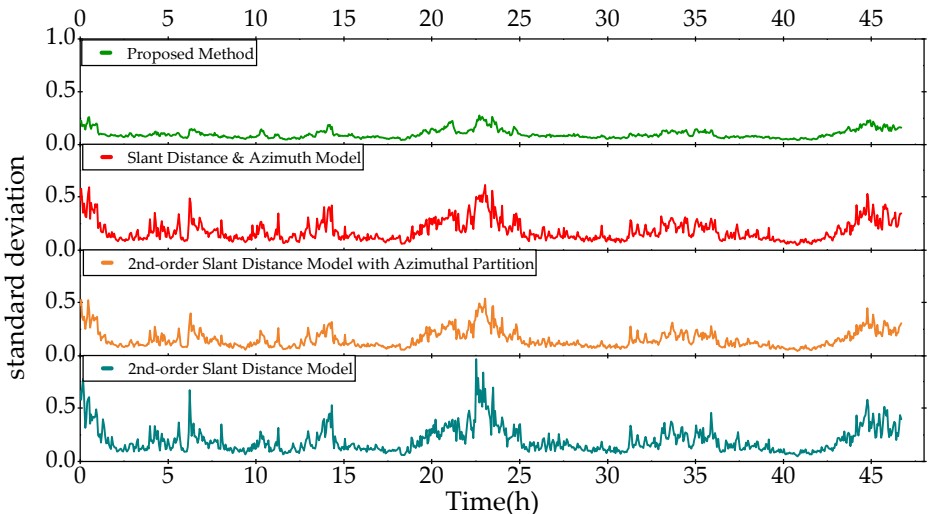

**Figure 15.** The variation curve of the residual AP's standard deviation after APC along the acquisition time.

**Table 3.** The statistical results of the residual AP's standard deviation after APC.

|  | Uncorrected | 2nd-Order Slant Distance Model | 2nd-Order Slant Distance Model with Azimuthal Partition | Slant Distance & Azimuth Model | Proposed Method |
| --- | --- | --- | --- | --- | --- |
| Mean Value | 0.2310 | 0.2071 | 0.1601 | 0.1897 | 0.1018 |
| Median Value | 0.1766 | 0.1631 | 0.1285 | 0.1523 | 0.0872 |

### 5.2. Simulation Experiment of Deformation Monitoring

To verify the tracking effects and retention rate of different methods on deformation, simulated deformation was added to some areas of the interferograms. The areas where deformation was added are shown in Figure 16, and none of the selected areas had obvious natural deformation. The added deformation is equivalent to 10 rad and was uniformly added to each interferogram. In order to reduce the effect of edge jump caused by partition, and the sudden phase change caused by noise, APC was performed by sliding average, i.e., the current interferogram and the previous $n - 1$ interferograms were added together to perform APC, and then the corrected phase value was divided by $n$, as the correction result of the current interferogram. In this experiment, $n = 10$. The algorithm parameters of the proposed method are shown in Table 2.

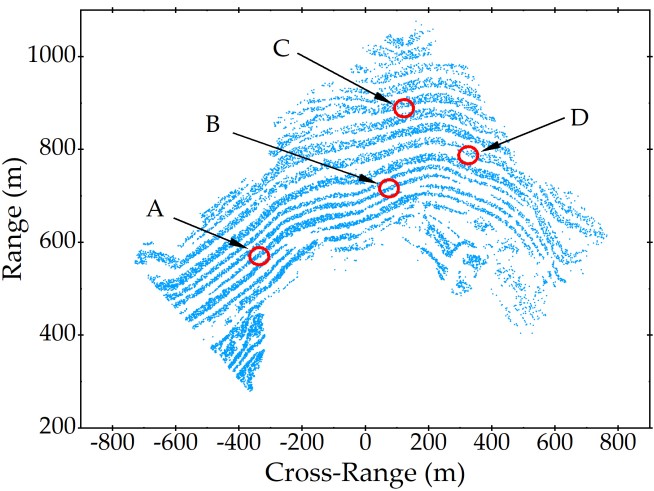

**Figure 16.** The areas where the deformation was added, A, B, C, and D mark the areas where deformation was added.

The median curves of the corrected CP for areas A, B, C, and D are shown as solid lines in Figure 17. Although the median curves' trends using the proposed method are similar to the simulated deformation curve, their slopes are slightly smaller, which indicates overfitting. In order to evaluate the degree of overfitting, the deformation retention rate (DRR) $n_{dr}$ was calculated as follows:

The relationship between a simulated deformation curve $D_s$ and time $t$ is:

$$D_s(t) = k_s t \tag{25}$$

where $k_s$ is the slope of the simulated deformation curve.

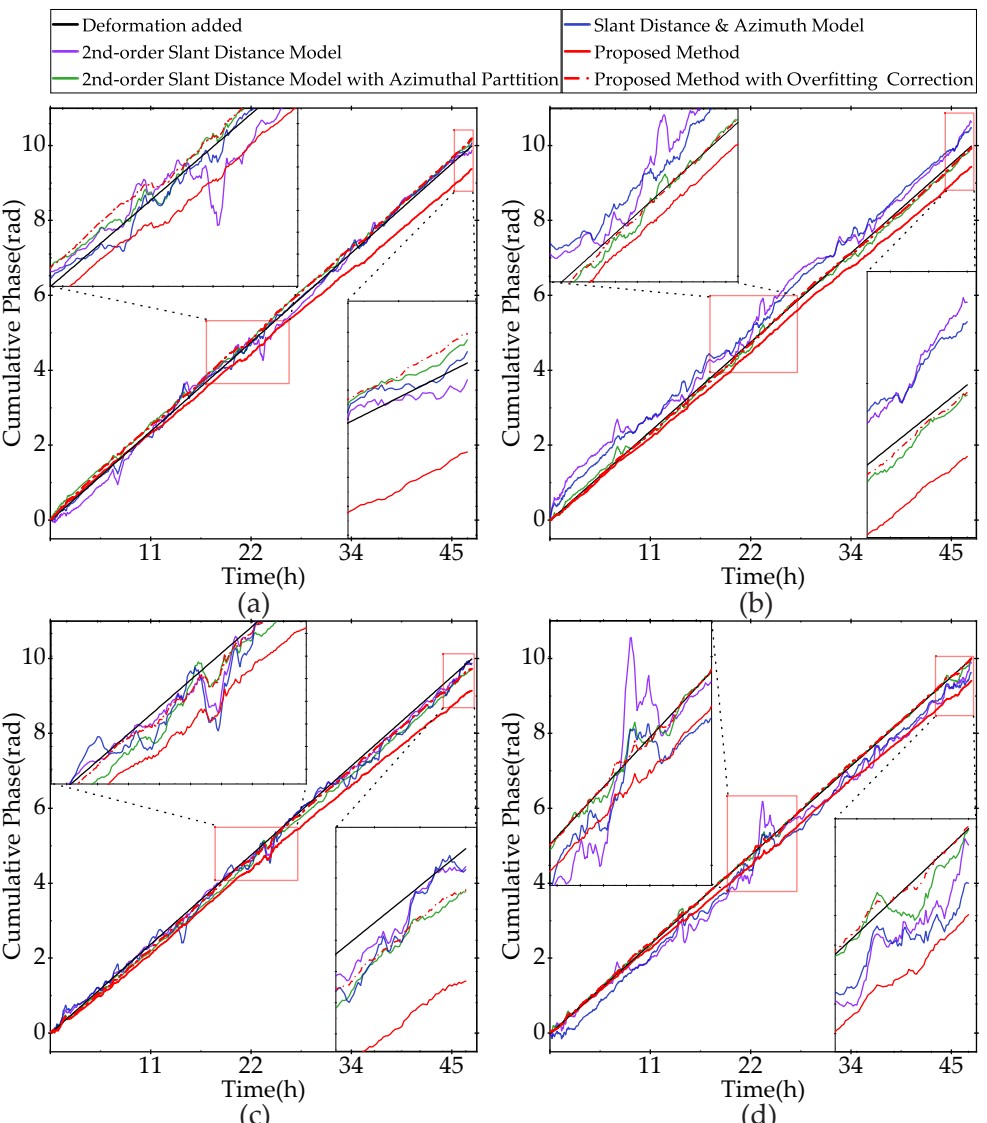

**Figure 17.** CP median curves of simulated deformation areas, (**a–d**) are the median curves of A–D in Figure 16, respectively.

The relationship between the corrected deformation curve $D_c$ and $t$ is:

$$D_c(t) = k_c t \tag{26}$$

where $k_c$ is the slope of the corrected deformation curve.

The least-squares method (LSM) was used to estimate $k_c$, based on the corrected deformation curves in A, B, C, and D. $n_{dr}$ is expressed as:

$$n_{dr} = \frac{k_c}{k_s} \tag{27}$$

Compared with calculating DRR based on the final cumulative phase value, estimating the slope of the cumulative phase curves by LSM uses more information and can better reflect the overall situation.

To attenuate the overfitting, an overfitting correction factor is introduced. The relationship between the overfitting-corrected phase $\varphi_O$ and the AP-corrected phase $\varphi_{AP}$, is shown in Equation (28).

$$\varphi_O = \frac{\varphi_{AP}}{n_{dr}} \tag{28}$$

In this experiment, $n_{dr} = 0.938$. With the change in algorithm parameters and monitoring situation, $n_{dr}$ also changes accordingly. To explore the effect of the algorithm parameters on DRR, a parameter setting analysis was conducted in Sections 5.3.2 and 5.3.3.

The overfitting-corrected curves are shown as red dotted lines in Figure 17. Calculating the difference between these curves and the simulated deformation curve, the result is shown in Figure 18. After APC, the CP curves and difference curves of the slant distance model and the slant distance & azimuth model have large fluctuations and high spikes during the period when the atmospheric conditions change more drastically, and the correction accuracy is poor. In contrast, the curves obtained by the proposed method are relatively flat, with lower fluctuations and spikes. Table 4 calculates the standard deviation of the phase difference curves in Figure 18. After overfitting correction, the proposed method has the smallest standard deviation and has a better tracking effect on the simulated deformation curve.

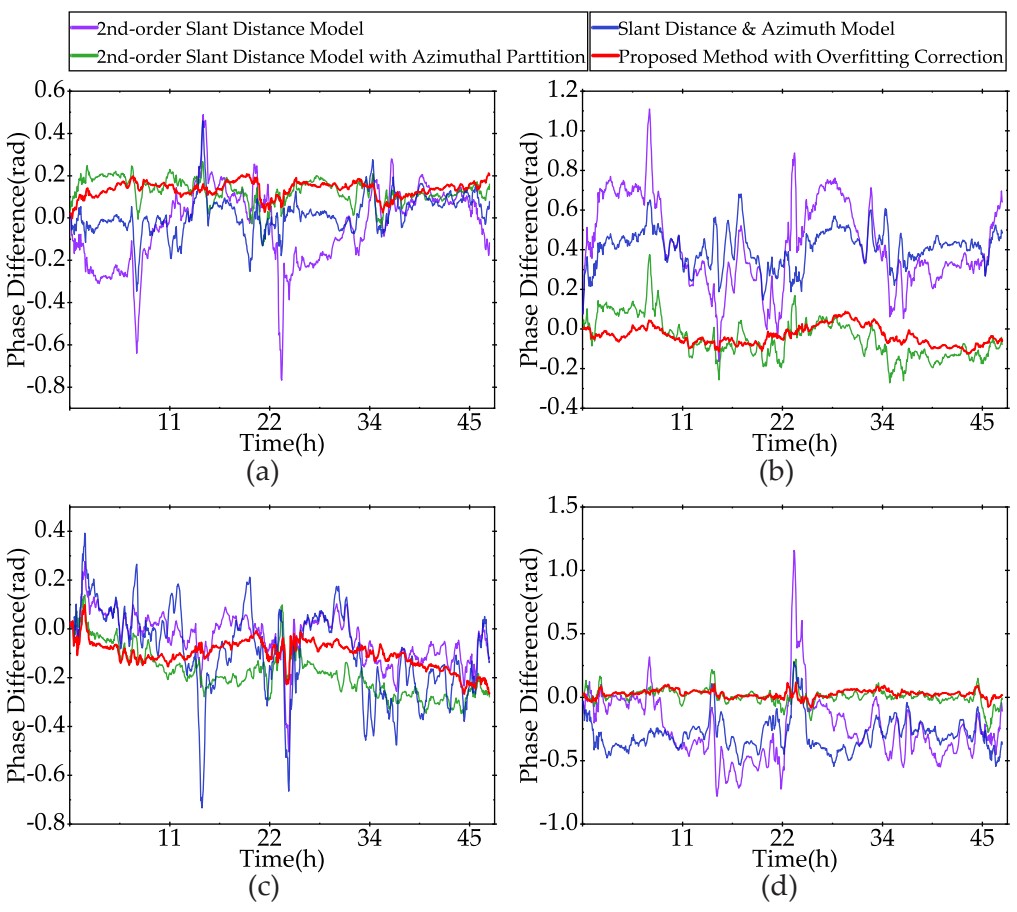

**Figure 18.** Phase difference curves between the median curves of CP for the four areas and the simulated deformation curve, (**a**–**d**) are the difference curves of A–D in Figure 16, respectively.

**Table 4.** The standard deviation of the phase difference curves in Figure 18.

|  | 2nd-Order Slant Distance Model | 2nd-Order Slant Distance Model with Azimuthal Partition | Slant Distance & Azimuth Model | Proposed Method with Overfitting Correction |
|---|---|---|---|---|
| A | 0.1818 | 0.5090 | 0.0910 | 0.0364 |
| B | 0.2170 | 0.1022 | 0.0941 | 0.0469 |
| C | 0.0969 | 0.0786 | 0.1791 | 0.0578 |
| D | 0.2567 | 0.0629 | 0.1278 | 0.0300 |

*5.3. Parameter Settings Analysis*

The proposed method has three important parameters: $k_{ph}$, $k_{cl}$, and $k_{nv}$. The effect of $k_{ph}$ and $k_{cl}$ on APC accuracy and DRR will first be analyzed. Then, appropriate values of $k_{ph}$ and $k_{cl}$ are selected, to discuss the effect of $k_{nv}$.

5.3.1. Effect of $k_{ph}$ and $k_{cl}$ on Atmospheric Phase Correction Accuracy

Set $k_{nv} = 100$, $k_{ph} = \{10, 20, 30, 40, 50, 60, 70, 80, 90, 100, 150, 200, 250, 300\}$, and $k_{cl} = \{5, 10, 15, 20, 25, 30\}$, to correct all interferograms. The standard deviation of the residual phase of each corrected interferogram was calculated separately, to obtain a standard deviation series, and then the mean value of the series was taken, which represents the overall accuracy. The smaller the mean value, the higher the accuracy. The statistical results of the mean values of standard deviation with different combinations of parameters, are shown in Figure 19.

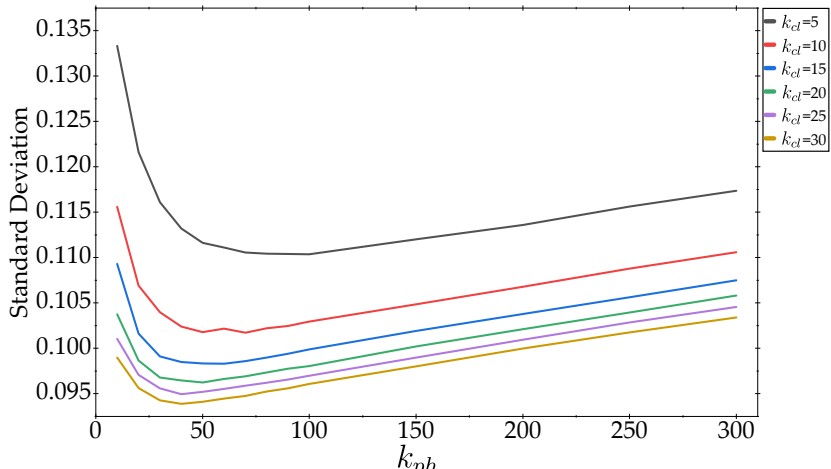

**Figure 19.** Mean values of images' standard deviation with different parameter combinations.

The larger $k_{cl}$ is, the smaller the sub-block is, so the APC accuracy improves with the increase in $k_{cl}$. However, the enhancement effect of $k_{cl}$ on accuracy decreases with its gradual increase. The APC accuracy presents a U-shaped relationship with $k_{ph}$. The accuracy first improves with the increase in $k_{ph}$, and the highest accuracy interval is $[25, 75]$. Then, the accuracy starts to decrease with the further increase in $k_{ph}$. When $k_{ph}$ is small, the $z$ component of the normal vector is close to 1, and when $k_{ph}$ is large, the $z$ component is close to 0. Both of these extreme cases lead to the concentrated distribution of the normal vector, that cannot be effectively distinguished by clustering. Only when the normal vectors are uniformly distributed, do the clustering results have the best discrimination.

5.3.2. Effect of $k_{ph}$ and $k_{cl}$ on Deformation Retention Rate

The settings of $k_{ph}$ and $k_{cl}$ are the same as in Section 5.3.1. Simulated deformation was added to all interferograms, with the same deformation parameter settings as in Section 5.2. Time series corrections were performed using different parameter combinations, and the DRR of cumulative deformation was subsequently counted, using the method in Section 5.2. The DRR curves are shown in Figure 20.

DRR decreases rapidly with the increase in $k_{cl}$, this indicates that the number of clusters has a great influence on overfitting. With the increase in $k_{ph}$, the DRR also decreases, but the rate of decrease is smaller. When $k_{ph}$ increases by a certain degree, the DRR will instead improve.

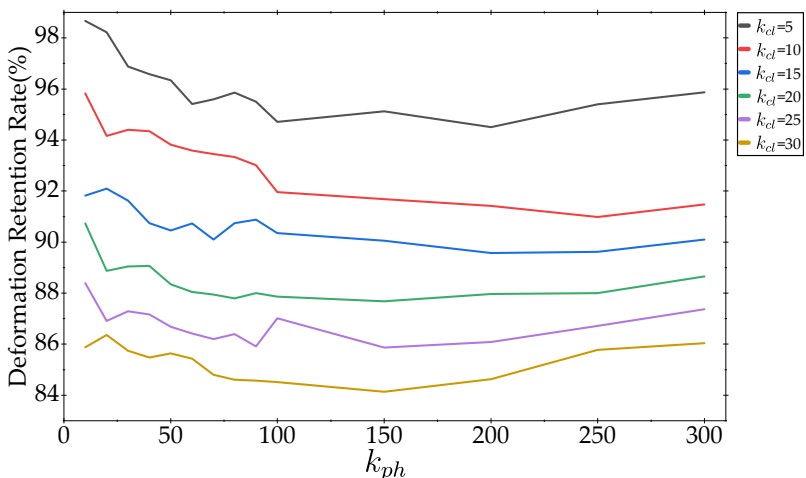

**Figure 20.** Deformation retention rate (DRR) curves under different parameter combinations.

### 5.3.3. Effect of $k_{nv}$

Considering the APC accuracy and DRR, we set $k_{ph} = 50$ and $k_{cl} = 10$, to analyze the effect of $k_{nv}$. The mean value of the standard deviation and DRR are shown in Figure 21. After the introduction of $k_{nv}$, the standard deviation decreases with the increase in $k_{nv}$, this finding suggests that $k_{nv}$ has a significant effect on the improvement of accuracy. However, this improvement diminishes as $k_{nv}$ increases. The DRR decreases with increasing $k_{nv}$, therefore, $k_{nv}$ cannot be increased unrestrictedly for the sake of higher accuracy.

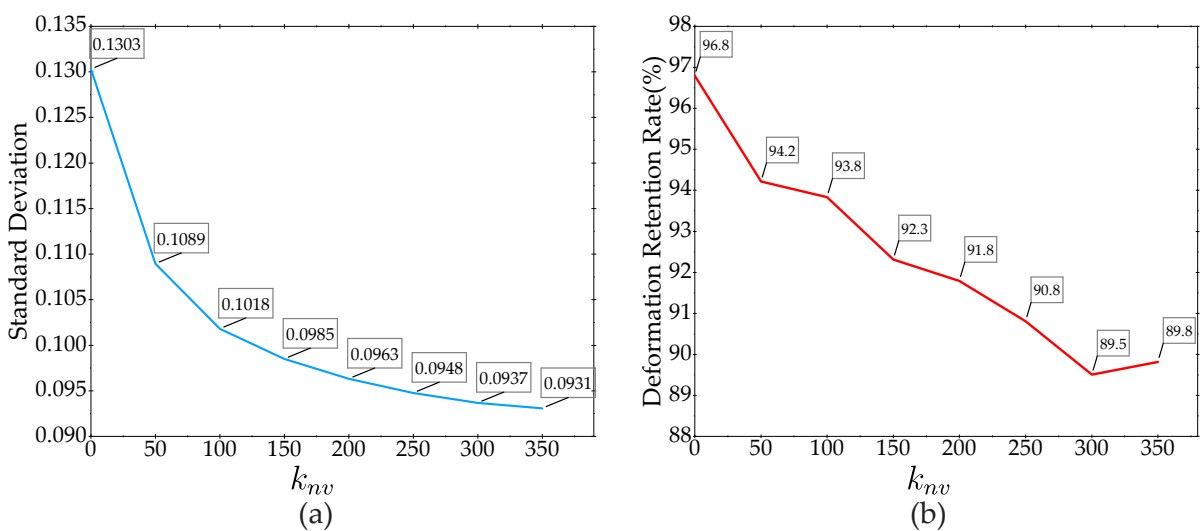

**Figure 21.** The mean value curve of standard deviation and DRR curves when $k_{ph} = 50$, $k_{cl} = 10$: (**a**) mean value curves of standard deviation, (**b**) DRR curve.

### 5.4. Time Series Atmospheric Phase Correction

The uncorrected CP curves for RPSs $p_1 - p_8$, and the corrected CP curves produced by the four methods, are shown in Figure 22. For the proposed methods, overfitting correction was performed after APC, and $n_{dr} = 0.938$. Due to the large residual AP, the curves of the 2nd-order slant distance model fluctuate most sharply. Its APC accuracy is improved after azimuthal partition, but the curves still have obvious fluctuations. In general, the APC accuracy of the slant distance & azimuth model is better than that of the slant distance model. After APC with the proposed method, the fluctuations of the curves at the RPSs are very small, mainly distributed between $-0.3$ and $0.3$ rad, and most of the AP is eliminated.

For the areas without obvious deformation, the proposed method achieved a better APC accuracy than the conventional methods.

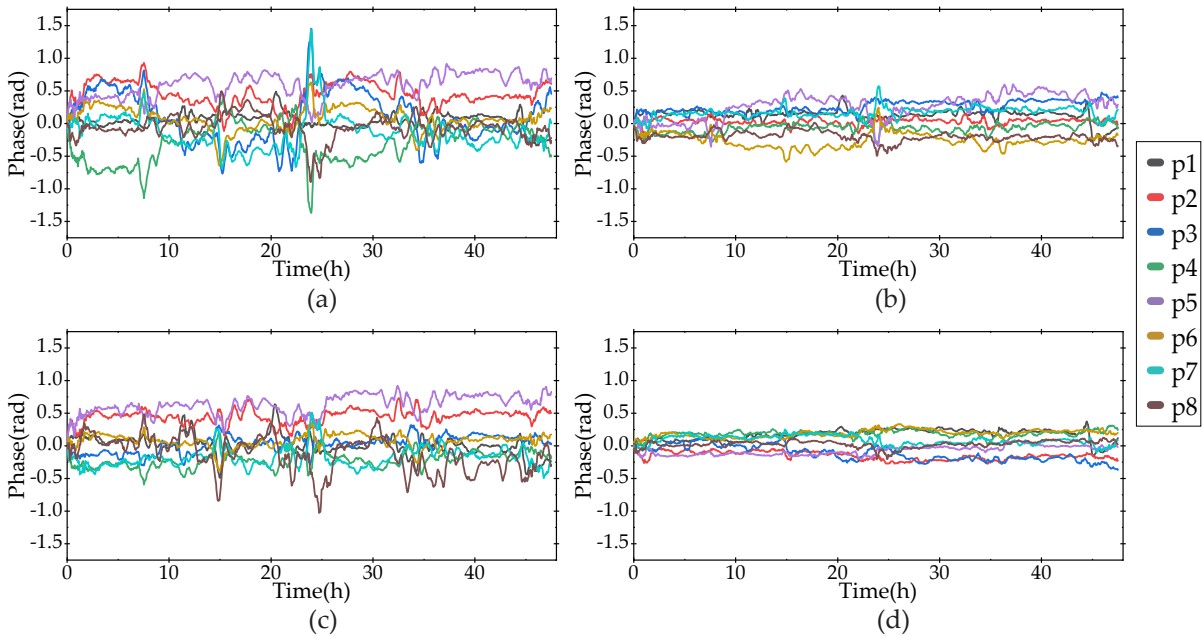

**Figure 22.** Corrected CP curves of RPSs $p_1$–$p_8$ in Figure 2: (**a**) 2nd-order slant distance model, (**b**) 2nd-order slant distance model with azumithal partition, (**c**) slant distance & azimuth model, (**d**) the proposed method with overfitting correction.

Figure 23 shows the final cumulative results of all interferograms corrected by the proposed method. A, B, and C, marked with red circles, are potential deformation areas, and their details are shown in Figure 24. The larger potential deformation areas in A, B, and C are marked with #1–#7, respectively.

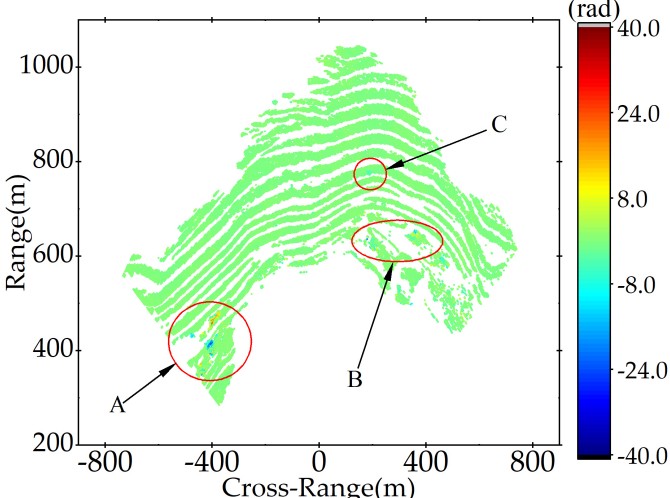

**Figure 23.** Final cumulative results after correction of all interferograms.

During the data acquisition, the bottom of the mine was being mined normally, and there were mining trucks in A and B, and A is the main operation area. Therefore, the phase changes in these two areas were mainly caused by mechanical operations and are not natural deformations. Among the seven marked small areas, #3, #4, and #7 have significant deformation, and their median curves of CP are shown in Figure 25.

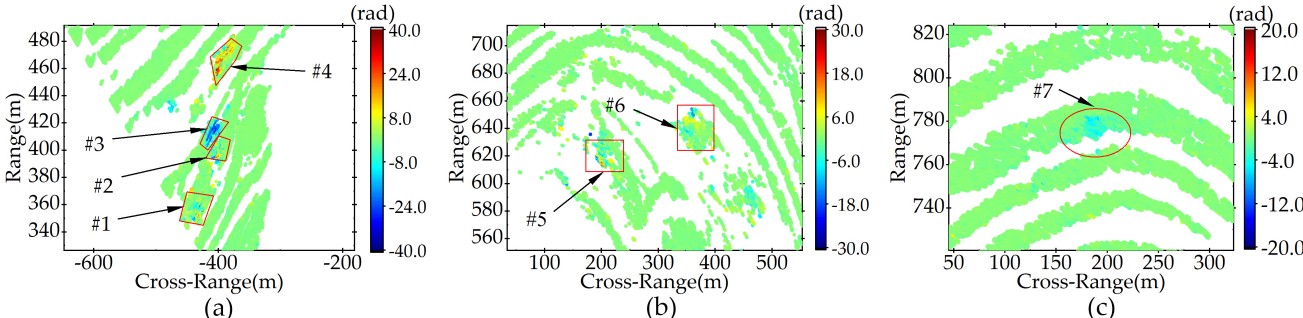

**Figure 24.** Details of (**a**) area A, (**b**) area B and (**c**) area C, the larger potential deformation areas are marked as #1–#7, respectively.

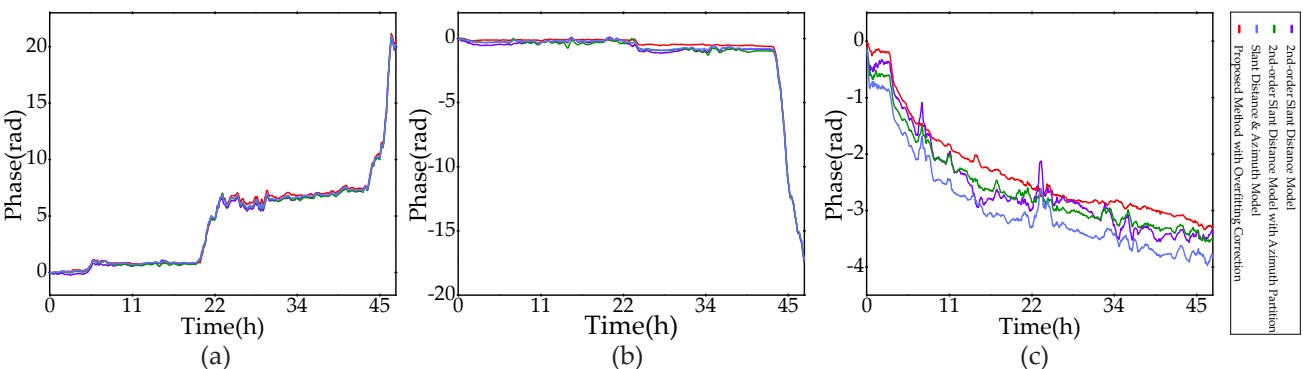

**Figure 25.** Corrected CP median curves of (**a**) #3, (**b**) #4, and (**c**) #7 in Figure 24 .

Figure 25, indicates that the curves of the conventional methods have obvious spikes in the interval with violent atmospheric changes, while the curves of the proposed method are smoother and consistent with the findings of the experiment in Section 5.2.

Both #3 and #4 are located in area A, in which mining vehicles operate, while small-scale blasting is conducted periodically, and the phase change in A is affected by both vehicle activity and mining-induced landslides. During the daytime, the mining activity is frequent and the CP curves are highly variable, while at night, when mining stops, the curves remain relatively stable. Area #7 is located in the middle of the mountain, far away from the operation area, without the interference of mining operations. The deformation area in #7 is small, and the CP curves are smooth overall and were slowly developing toward the negative direction in both the daytime and nighttime, so this can be considered as a naturally occurring deformation. For #7, the corrected deformation of the proposed method is generally smaller than that of the conventional methods, which is mainly because the atmospheric variation was large during the beginning of the acquisition period, making the curves of the conventional methods jump downward, while the proposed method is highly accurate and thus the jump is not obvious.

## 6. Discussions

### 6.1. Complicated Distribution of Atmospheric Phase

The interferogram in Figure 3 shows spatial phase wrapping with significant AP, while the interferogram in Figure 11 illustrates the complicated distribution of AP in different directions. From the time-series CP curves of the RPSs in Figure 4, and the standard deviation curves representing the complexity of the AP distribution in Figure 5, the AP fluctuate greatly with time and there is obvious periodicity. These indicate that in a real monitoring scenario, multiple factors interact with each other and make the APS exhibit strong inhomogeneity in both space and time, seriously affecting the accuracy of deformation measurement. In this research, the factors affecting the AP include:

(1) High altitude. The altitude of the Dabao Mountain is between 600 and 800 m. The solar radiation is stronger than that in lower altitude areas, and the air is relatively thin and poorly insulated. This also leads to large changes in atmospheric parameters within a short period of time, and therefore significant diurnal variations. During the daytime, the monitored area is mostly in clear or cloudy weather, and there was no obvious rainfall process during the data acquisition, so the influence of solar irradiation on temperature and humidity is significant. The heat brought by the sun makes the atmospheric parameters change significantly, so the distribution of the AP is more dispersed, and the spatial phase wrapping phenomenon can easily occur. After the rapid loss of heat at night, the change in atmospheric parameters tends to be smooth, and the AP becomes smaller and more stable accordingly.

(2) Steep terrain of the mine. The Dabao Mountain Mine has been mined for many years and the mountain is very steep. The relative elevation from the bottom of the pit to the top of the mine is about 150 m. The difference in elevation makes the spatial distribution of atmospheric parameters non-uniform, resulting in significant changes in the spatial distribution of AP with the change in elevation.

### 6.2. Comparison of the Conventional Methods and the Proposed Method

The information used to fit the regression coefficients in the conventional methods includes slant distance, azimuth, and elevation, which are not sufficient to reflect the spatial distribution pattern of APS comprehensively. The order of the model is generally first or second order, with which it is difficult to simulate the complicated APS. For scenarios with good atmospheric homogeneity, such conventional methods can achieve good correction, but for APS with complicated spatial distribution, their APC accuracy is not satisfactory. As can be observed from Figure 12, there are large AP residuals in both the range and cross-range direction after APC, and the CP curves of the conventional methods all have obvious spikes. Partition in the azimuth direction effectively reduces residual AP, and the CP curves' fluctuations of the 2nd-order slant distance model, indicating that partition methods can significantly improve APC accuracy.

The proposed method divides the sub-blocks according to the spatial distribution of APS, the close areas with the same distribution are divided into a sub-block for correction, which ensures the APC accuracy and avoids the occurrence of serious overfitting at the same time. Compared to conventional methods, the proposed method has significantly improved correction results. As shown in Figure 13, the residual AP of the corrected interferogram in both directions is concentrated around 0. As shown in Figure 15, the standard deviation of the corrected residual AP is also the smallest among the results of all four methods. Both for simulated deformation and real deformation, the CP curves obtained by the proposed method are smoother in Figures 13, 17 and 25, indicating a higher APC accuracy.

Since the estimation of the APS is performed in sub-blocks, without considering the block-to-block relationship, the estimated APS may have a phase jump at the edge of the sub-blocks. To reduce the effect of APS discontinuities during long-term monitoring, a sliding average correction method is used. The results show that the cumulative curves of the proposed method do not show significant jumps, indicating that the phase discontinuity has almost no effect on the long-term correction.

### 6.3. Conflict between Accuracy and Credibility

The deformation component in the interference phase will inevitably affect the APC, which leads to the overfitting problem. There is a conflict between correction accuracy and reliability when using the block method for APC. Increasing the number of blocks can improve the estimation ability of the correction method for APS, reduce the residual atmospheric phase after correction, and thus improve the accuracy of correction. If the block is too small, the proportion of deformation part in it will be too large, which will have a significant impact on the parameter estimation of APC, leading to overfitting.

GB-SAR systems are often applied in important fields, such as disaster monitoring. Compared with false alarms, missing reports may lead to premature warnings and disastrous consequences. Therefore, it is necessary to avoid the excessive requirement of precision and lead to a large number of deformation losses. Both accuracy and DRR should be taken into consideration when setting parameters. The chosen parameters have a great influence on the correction accuracy and DRR. Hence, different parameters are selected for experiments in this paper, to analyze the influence of the parameters on the APC results and guide the selection of the parameters. It is worth noting that the universality of the parameter analysis results has not been verified, due to the lack of sufficient datasets. The relevant literature generally pays more attention to the accuracy, rather than overfitting and DRR. Consequently, systematic theories or empirical formulas have not been formed yet, which needs to be further studied.

## 7. Conclusions

This article focuses on a high-accuracy APC method for GB-SAR, based on normal vector clustering partition. Through the analysis of the data, it is found that the atmosphere changes drastically in complicated conditions. Even when the time baseline is very short, there are still a large amount of AP in the interferometric phases. In some cases, AP lead to spatial phase unwrapping. For complicated APS, the residual AP is still very obvious after correction with conventional regression models.

The proposed method combines the spatial normal vectors of local APS, with the position coordinates of PSs as the dataset, and uses a clustering approach to partition the APS according to its distribution, followed by the partitioning of APC.

APC and simulation experiments on measurement data, suggest that the proposed method achieves higher accuracy than the conventional model-based correction methods, for APS under complicated conditions with drastic changes. At the same time, due to the introduction of a spatial normal vector in the process of partitioning, the partitioning results are consistent with the distribution of APS. The number of sub-blocks is minimized, thus avoiding severe overfitting, and the deformation component is preserved as much as possible, realizing the balance between accuracy and credibility. This article verifies the feasibility and effectiveness of using APS distribution information to guide the block and conduct APC, and provides a new idea for further research.

The analysis of the algorithm parameters demonstrates that the parameter settings have a significant impact on the APC accuracy and deformation retention rate, and that they should be set reasonably. However, due to the lack of more data for verification, a systematic parameter selection theory or empirical formula has not been formed, which needs further research. At present, researchers usually pay more attention to the accuracy, and pay less attention to overfitting and deformation retention rate, and relevant studies need to be strengthened.

**Author Contributions:** Conceptualization, P.O. and T.L.; methodology, P.O. and T.L.; software, P.O. and S.H.; validation, P.O., S.H., W.C. and D.W.; formal analysis, P.O., W.C. and D.W.; investigation, P.O. and T.L.; resources, P.O. and T.L.; data curation, P.O.; writing—original draft preparation, P.O.; writing—review and editing, T.L.; visualization, S.H., W.C. and D.W.; supervision, T.L., S.H. and W.C.; project administration, T.L.; funding acquisition, T.L. All authors have read and agreed to the published version of the manuscript.

**Funding:** This research was funded by Key Areas of R&D Projects in Guangdong Province (2019B111101001), the Shenzhen Science Technology Planning Project (JCYJ20190807153416984), and the Natural Science Foundation of China (62071499).

**Data Availability Statement:** The data presented in this study are available from the corresponding author upon request.

**Conflicts of Interest:** The authors declare no conflict of interest.

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
