# Peer review of "An Atmospheric Phase Correction Method Based on Normal Vector Clustering Partition in Complicated Conditions for GB-SAR"

_remotesensing, doi:10.3390/rs15071744_

Round 1

Reviewer 1 Report

In this paper, aiming at the problem of eliminating complex atmospheric phase in the application of ground-based synthetic aperture radar (GB-SAR), a GB-SAR high-precision APC method based on normal vector clustering partition is studied on the basis of three commonly used simulation models. Through the radar experiments on Dabao Mountain Mine, it is proved that this method can produce lower residual atmospheric phase and cumulative phase curves fluctuations, and can effectively retain the deformation phase. The effectiveness of the method in tracking deformation and correction accuracy is verified by comparative experiments.

But there are some questions in this paper which need you to notice and answer.

1. For spatial phase wrapping phenomenon appeared, how do you unwrap and do you deal with the unwrapping error? please describe the steps.

2. In section 3.2.1 (line 217), there is a misspelling (A wxample of the original PSs and the corresponding CPSs are shown in Figure 7) that should be ‘example’ not ‘wxample’.

3. There is no occurrence of parameter  in Formula 10, which is explained on line 178. This explanation should be put after equation 11.

4. In section 4.1, why do you choose to show the correction results of the seventh interference image and how does it differ from other interference images?

5. In section 4.2, in order to reduce the effect of overfitting, overfitting correction factor  is introduced, can you explain in detail why the median value of the cumulative phase in all simulated deformation areas was selected?

6. The lines in Figure 17 which drawn the cumulative phase median curves of simulated deformation areas are too clustered, which is a little difficult to observe, perhaps improvements are needed to make them clearer.

Reviewer 2 Report

The topic of the work is certainly of interest to specialists in the GBSAR technique, and the paper is well written and well presented.

However, I have some major concerns regarding the applicability and effectiveness of the proposed method.  I believe that the work can be published after having better discussed these aspects.

Below, my observations:

Table1: the symbols > and < are not appropriate for technical specifications. For example, if the maximum detection range is 4000 m please just write 4000 m, otherwise if the maximum detection range is greater than 4000 m please write its actual value. Please replace the > and < symbols with the actual specification values.

Equation 6: To my knowledge the wet component proportional to 1/T^2, doesn’t depend on pressure but only on humidity and temperature. Please check this formula.

Line 178: after equation 10 the azimuth angle is defined, but azimuth angle is introduced in equation 11. Please move the azimuth angle definition after equation 11.

Line 224: The phase amplification procedure seems quite arbitrary and scenario dependent, moreover the clustering results will depend heavily on how phase is amplified. Is there a way to make this procedure more automatic and less arbitrary? Please discuss this aspect in more detail.

Line 255: How is defined the parameter ‘m’? Is it maybe ‘k’?

Line 263: ‘k’ seems a very important parameter and, again it seems quite arbitrary. How can you select it in an automatic way? Please discuss this aspect in more detail.

Line 275: data instead of date

Line 287: can you better explain what do you mean by ‘retention and tracking effect of deformation’?

Line 348: ‘k’ is already used to identify the number of clusters. Please use a different symbol to avoid confusion.

Line 359: In my opinion, overfitting correction seems very arbitrary, and in a real scenario case it is not possible to assume that there is simply a factor 0.94 to be used to correct the results.

Figure 17 caption: (d) instead of (b)

Figure 18 caption: (d) instead of (b)

Reviewer 3 Report

This paper proposed a clustering partition method based on the normal vector of APS, which can partition the complicated APS more reasonably, and then correct the AP based on the partition results. Experiments show that the proposed method can be well adapted to complicated atmospheric conditions and can effectively improve the deformation monitoring accuracy. It can be accepted after revising these concerns:

1.         The novelty and motivation of this work is not very clear. Please rewrite the abstract and introduction to show the motivations and contributions of this work.

2.         Avoid lumping references as in [x-y], [x, y] and all other. It is not necessary to give several references that say exactly the same.

3.         The typesetting needs to be improved.

4.         The third section should mainly introduce the innovative work of this article, so whether it is necessary to introduce the Conventional Methods in this part. If it is necessary, whether it is more appropriate to put it in the part of such as related work.

5.         The main innovation of this article in Figure 6 is not obvious.

6.         Are the images and their ground truth freely available? How about the results on other datasets?

7.         It seems necessary to conduct parameter setting analysis or sensitivity analysis.

8.         Make sure your conclusions reflect on the strengths and weaknesses of your work, how others in the field can benefit from it and thoroughly discus future work.

9.         The format of references needs to be modified.

10.     The writing of the paper needs to be further polished to make it publishable.

Round 2

Reviewer 2 Report

I think that the authors have exhaustively answered to my observations and therefore the work can be accepted in the present form

Reviewer 3 Report

It seems that the authors have addressed all the requests.